# Ceftolozane/Tazobactam and Ceftazidime/Avibactam for Multidrug-Resistant Gram-Negative Infections in Immunocompetent Patients: A Single-Center Retrospective Study

**DOI:** 10.3390/antibiotics9100640

**Published:** 2020-09-24

**Authors:** Rosario Cultrera, Marco Libanore, Agostino Barozzi, Erica d’Anchera, Letizia Romanini, Fabio Fabbian, Francesco De Motoli, Brunella Quarta, Armando Stefanati, Niccolò Bolognesi, Giovanni Gabutti

**Affiliations:** 1Infectious and Tropical Diseases, Department of Morphology, Surgery and Experimental Medicine, University ‘S. Anna’ Hospital of Ferrara, Via Aldo Moro 8, 44124 Ferrara, Italy; ctr@unife.it; 2Infectious Diseases Unit, University ‘S. Anna’ Hospital of Ferrara, Via Aldo Moro 8, 44124 Ferrara, Italy; m.libanore@ospfe.it; 3Clinical Microbiology, University ‘S. Anna’ Hospital of Ferrara, Via Aldo Moro 8, 44124 Ferrara, Italy; a.barozzi@ospfe.it (A.B.); l.romanini@ospfe.it (L.R.); 4Postgraduate School of Hygiene and Preventive Medicine, University of Ferrara, Via Fossato Di Mortara 64/B, 44121 Ferrara, Italy; dncrce@unife.it (E.d.); dmtfnc@unife.it (F.D.M.); blgncl@unife.it (N.B.); 5Clinica Medica Unit, Department of Medical Sciences, University of Ferrara, Via Aldo Moro 8, 44124 Ferrara, Italy; fbbfba@unife.it; 6Department of Pharmacy, University ‘S. Anna’ Hospital of Ferrara, Via Aldo Moro 8, 44124 Ferrara, Italy; b.quarta@ospfe.it; 7Public Health Medicine Section, Department of Medical Sciences, University of Ferrara, Via Fossato di Mortara 64/B, 44121 Ferrara, Italy; sta@unife.it

**Keywords:** carbapenem-sparing regimen, ceftazidime/avibactam, ceftolozane-tazobactam, ESBL-producing *Enterobacterales*, healthcare-associated infections

## Abstract

Complicated infections from multidrug-resistant Gram-negative bacteria (MDR-GNB) represent a serious problem presenting many challenges. Resistance to many classes of antibiotics reduces the probability of an adequate empirical treatment, with unfavorable consequences, increasing morbidity and mortality. Readily available patient medical history and updated information about the local microbiological epidemiology remain critical for defining the baseline risk of MDR-GNB infections and guiding empirical treatment choices, with the aim of avoiding both undertreatment and overtreatment. There are few literature data that report real-life experiences in the use of ceftolozane/tazobactam and ceftazidime/avibactam, with particular reference to microbiological cure. Some studies reported experiences for the treatment of MDR-GNB infections in patients with hematological malignancies or specifically in *Pseudomonas aeruginosa* infections. We report our clinical single-center experience regarding the real-life use of ceftolozane/tazobactam and ceftazidime/avibactam to treat serious and complicated infections due to MDR-GNB and carbapenem-resistant *Enterobacterales* (CRE), with particular regard given to intra-abdominal and urinary tract infections and sepsis.

## 1. Introduction

Infections due to multidrug-resistant (MDR) Gram-negative bacteria (GNB) are difficult to treat, representing an actual serious health emergency, particularly in patients who often have comorbidities. These difficult infections are responsible for high direct costs consequent to the use of new antimicrobial drugs and the indirect costs of prolonged hospitalization and healthcare-related expenditure [1,2]. The increase in infections by *Pseudomonas aeruginosa, Acinetobacter baumannii, Klebsiella pneumoniae,* and extended-spectrum β-lactamase-producing (ESBL) or carbapenem-resistant *Enterobacterales* (CRE) have hindered the treatment of these infections, with a consequent increase in morbidity and mortality [3,4,5,6,7].

The link between an increased consumption of carbapenems, considered as last-resort antibiotics for the treatment of infections due to MDR Gram-negative bacteria, and the emergence of carbapenemases-producing *Enterobacterales* (CPE) other than CRE is now well-demonstrated [8,9].

The need for new and effective anti-CRE therapies, together with an adequate infection source control practice, is urgent. Currently, antibiotic options include high doses and strategies of combination therapies with polymyxins, tigecycline, fosfomycin, or aminoglycosides to maximize treatment success. Moreover, the new β-lactam/β-lactamase inhibitors should be considered in severe CRE infections [10]. Ceftazidime/avibactam has recently been approved for the treatment of complicated intraabdominal infections (cIAIs), complicated urinary tract infections (cUTIs), and hospital-acquired and ventilator-associated bacterial pneumonia (HABP and VABP). The European Medicine Agency (EMA) authorized the use of ceftazidime/avibactam also in adult patients with infections due to aerobic GNB with limited therapeutic options. Unlike most β-lactamase inhibitors, avibactam is a novel synthetic non-β-lactam (diazabicyclooctane)/β-lactamase inhibitor that inhibits a wide range of β-lactamases, including Ambler Class A (GEM, SHV, CTX-M, and KPC), Class C (AmpC), and some Class D (OXA-48) β-lactamases [11], expanding the activity spectrum of ceftazidime to MDR Gram-negative bacteria. It does not inhibit Class B MBLs (IMP, VIM, VEB, and NDM) [12,13].

Ceftolozane/tazobactam has recently been approved for the treatment of cIAIs, cUTIs, and HABP and VABP. Tazobactam protects ceftolozane against ESBLs *Enterobacterales*, as demonstrated by Phase III trials [14]. Ceftolozane is notably active against *P. aeruginosa*, with minimum inhibitory concentrations (MICs) lower than those of ceftazidime, one of the most active anti-pseudomonal β-lactams. This activity is retained for many strains with derepressed AmpC or up-regulated efflux [15].

Here, we report the clinical experience with ceftazidime/avibactam and ceftolozane/tazobactam to treat or consolidate the treatment of serious Gram-negative infections in the University Hospital of Ferrara, Italy, between 2017 and 2019. The primary objective was to describe the microbiological cure in the study population. The secondary objectives were to describe (i) the appropriateness of ceftolozane/tazobactam and ceftazidime/avibactam antibiotics in GNB infections; (ii) the adverse events related to ceftazidime/avibactam and ceftolozane/tazobactam treatment; (iii) the 30-day mortality.

## 2. Results

### 2.1. Group Treated by Ceftolozane/Tazobactam

A total of 241 microbiological isolates obtained from 122 consecutive patients treated with ≥72 h of ceftolozane/tazobactam for GNB infections were included in the study. The median age of patients was 64 years and 59% (72/122) were male. As many as 109/241 samples (45.2%) were obtained from peritoneal fluid, 58/241 (24.1%) from urine, and 48/241 (19.9%) from blood samples. A total of 26/241 samples (10.8%) were obtained from different biological samples (synovial fluid, bone, surgical wounds) from patients with other conditions (bone infections, surgical site infections, septic arthritis, skin and soft tissue infections) (Table A1). The isolated bacteria causing the most serious infections were *P. aeruginosa* (n = 69; 28.6%), *K. pneumoniae* (n = 54; 22.4%), *E. coli* (n = 35; 14.5%), *Serratia marcescens* (n = 13; 5.4%), and *A. baumannii* (n = 3; 1.2%) (Figure A1). ESBL-producing *Enterobacterales* were 136/241 (56.4%).

Eighty-eight patients (72%) received ceftolozane/tazobactam as a second-line therapy, with the median time for switching to ceftolozane/tazobactam as 8 days (interquartile range (IQR) 3–10 days). Piperacillin/tazobactam (n = 35), meropenem (n = 14), amoxicillin-clavulanic acid (n = 11), and tigecycline (n = 8) were the most common antimicrobials prescribed prior to the initiation of treatment by ceftolozane/tazobactam. The switch to ceftolozane/tazobactam was mainly due to the documented resistance to previous antibiotics and the lack of clinical response to previous antibiotic treatment. In some cases, patients received ceftolozane/tazobactam treatment in combination with other antibiotics, mainly targeting possible Gram-positive microorganisms, including tigecycline (n = 5), daptomycin (n = 4), vancomycin (n = 3), teicoplanin (n = 3), and linezolid (n = 3). *Enterococcus faecalis* and *E. faecium* were isolated in three patients with cIAIs and cUTIs. Five patients received empirical combined ceftolozane/tazobactam treatment, supposing possible Gram-positive bacteria pneumonia or polymicrobial infections. Of these, polymicrobial Gram-negative infections were shown in 24 patients treated with ceftolozane/tazobactam. The 30-day all-cause mortality was 20.5% (25/122). Microbiological analyses on blood, urine, or peritoneal fluid samples obtained at 48 h and 72 h and 7 days after the discontinuation of antibiotic therapy were performed. A microbiological cure was shown in 99/122 (81%) patients, in 81.1% of the patients treated with ceftolozane/tazobactam alone, and in 78.3% of those treated with combination therapy. Particularly, microbiological negativity was shown in 94.4%, 77.4%, and 95.8% for isolates of ESBL-producing *P. aeruginosa*, *K. pneumoniae*, and *E. coli*, respectively. The usefulness of ceftolozane/tazobactam in real practice was evidenced in infections due to ESBL-producing *Enterobacterales* (*p* < 0.001) and in meropenem-sensitive infections (*p* < 0.001) compared with meropenem-resistant infections. Moreover, these results confirmed a possible role of ceftolozane/tazobactam in a carbapenem-sparing policy in the context of an antibiotic stewardship. The appropriateness was confirmed by the use of ceftolozane/tazobactam in infections due to GNB resistant to third-generation cephalosporins and fluoroquinolones, according to specific indications approved by our local regulatory organizations (Table A2). No adverse events related to ceftolozane/tazobactam were observed in the study population.

### 2.2. Group Treated by Ceftazidime/Avibactam

A total of 125 microbiological isolates were obtained from 47 patients treated with ≥72 h of ceftazidime/avibactam for GNB infections. Their median age was 62.2 years (73.8% male). As many as 70/125 samples (56%) were obtained from patients with sepsis and septic shock, 40/125 (32%) from patients with cUTIs, 12/125 samples (9.6%) were obtained from patients with cIAIs, and 3/125 (2.4%) from patients with pneumonia (Table A1). The main microbiological isolates were *K. pneumonia* (n = 55; 44%), *E. coli* (n = 15; 12%), and *P. aeruginosa* (n = 14; 11.2%) (Figure A2). Extended-spectrum β-lactamase (ESBL)-producing *Enterobacterales* were 63/125 (50.4%). Twenty-five patients (53.2%) received ceftazidime/avibactam as a second-line therapy, with the median time for switching to ceftazidime/avibactam as 5 days (interquartile range (IQR) 3–8 days). Meropenem (n = 6), piperacillin/tazobactam (n = 5), tigecycline (n = 5), ceftriaxone (n = 2), and ceftolozane/tazobactam (n = 2) were the most common antimicrobials prescribed prior to the initiation of treatment with ceftazidime/avibactam. The switch to ceftazidime/avibactam was mainly due to the documented resistance of microbiological isolates to antimicrobial susceptibility tests and the lack of clinical response to previous antibiotic treatment.

Ceftazidime/avibactam was administered as a combination treatment in 24 patients (51.1%), and the median length of therapy was 12 days. The antimicrobial agents most frequently used in combination with ceftazidime/avibactam included meropenem (N = 5), tigecycline (N = 4), daptomycin (N = 3), and vancomycin (N = 3). *Enterococcus faecalis* and *E. faecium* were isolated in four patients with cIAIs and cUTIs. Polymicrobial Gram-negative infections were shown in 13 patients treated with ceftazidime/avibactam.

Among the 47 patients treated with ceftazidime/avibactam, the 30-day all-cause mortality was 31.9% (15/47). Microbiological analyses on blood, urine, or peritoneal fluid samples obtained at 48 h 72 h and 7 days after the discontinuation of antibiotic therapy were conducted. A microbiological cure was shown in 31/47 (65.9%) patients, in 67.6% of patients treated with ceftazidime/avibactam alone, and in 69.9% of those treated with combination therapy. A good microbiological cure was shown in all cases where ceftazidime/avibactam was combined with meropenem or daptomycin. Particularly, microbiological negativity was shown in 75% and 53.3%, for isolates of ESBL-producing *K. pneumoniae* and *E. coli*, respectively. These results evidenced the usefulness of ceftazidime/avibactam in real practice in carbapenem-resistant GNB infections compared with carbapenem-sensitive GNB infections, as showed by our evidence of *E. coli* and *K. pneumoniae* with a low sensitivity to meropenem (52.4%; *p* < 0.05), ertapenem (45.7%; *p* < 0.001), and ceftazidime (5.7%; *p* < 0.001). The appropriateness of ceftazidime/avibactam was confirmed by its use in infections due to carbapenem-resistant bacteria, according to specific indications defined by our regulatory organizations (Table A3).

No adverse events related to ceftazidime/avibactam were observed in the study population.

## 3. Discussion

Healthcare-associated infections from GNB are very frequent, and those caused by MDR, XDR, and PDR microorganisms are increasing. Among the infections that do not fall into this classification, those recently defined as DTR are increasingly relevant, representing the real condition of limited availability of first-line drugs in antibiotic therapy. Moreover, empirical therapeutic options are often not adequate for factors such as pharmacokinetic and pharmacodynamic properties and for their toxicity. Tigecycline, fluoroquinolones, and aminoglycosides were described to be the most frequently used antibiotics [16,17]. GNB infections with recognized DTR resistance are related to a high mortality. This creates the need for effective antibiotics with, at the same time, a good tolerability even more urgent. The new combination of cephalosporins/β-lactamase inhibitors can also be considered a resource in the treatment of infections due to DTR microorganisms. This retrospective single-center study evaluated a cohort of patients treated with two new associations of β-lactam/β-lactamase, ceftolozane/tazobactam, and ceftazidime/avibactam for infections due to ESBL-producing and carbapenemases-producing *Enterobacterales*. In particular, the microbiological response to therapy with these two drugs and the appropriateness of their use on the basis of the determined sensitivities of the isolated microorganisms were studied.

The increase in infections due to ESBL-positive *Enterobacterales* induced an increased use of β-lactams and penicillins associated with β-lactamase inhibitors and, in particular, piperacillin/tazobactam. During the period considered in our study, the increased demand for piperacillin/tazobactam was followed by some periods of unavailability of this antibiotic.

Another determining factor in this choice was the containment of the use of meropenem with a view to a carbapenem-sparing policy, also considering the percentage of sensitivity found in *P. aeruginosa* to meropenem.

A good response to therapy has been observed in patients treated with ceftolozane/tazobactam, demonstrated by the high number of negative samples. The high efficacy of ceftolozane/tazobactam in infections with *P. aeruginosa* but also in infections with ESBL-producing *Enterobacterales* has been confirmed. The study highlighted the use of ceftolozane/tazobactam also in patients with ESBL-non-producing GNB infections. A careful analysis of the data has shown that this use coincided temporally with the lack of availability of piperacillin/tazobactam in Italy due to the high demand due to the increase in the incidence of ESBL-producing *Enterobacterales* [18].

Considering the different clinic syndromes included in our study and limited to the few cases described, a particular consideration could be made about patients with cIAIs (58/122) treated with ceftolozane/tazobactam. A total of 7/58 received ceftolozane/tazobactam as a first-line therapy. When it was used as a second-line therapy, the antibiotics more used were piperacillin/tazobactam (19/58), tigecycline (7/58), amoxicillin/clavulanic acid (8/58), and meropenem (6/58). In these patients, we observed a rapid response to ceftolozane/tazobactam therapy, with the microbiological cure time within 7 days of treatment. Biological samples from patients were collected at 48 h and 72 h and 7 days after the end of therapies in all patients as the antibiotic stewardship program of our Hospital. Follow-up was made by cultures of urine and blood samples for cUTIs, blood, and peritoneal fluid samples for cIAIs, blood samples for sepsis and bloodstream infections, and bronchoalveolar lavage or sputum for respiratory tract infections. Therapy duration was defined at 7–10 days for cIAIs, 7–14 days for bloodstream infections, 10–14 days for cUTIs, and 5–7 days for CAP and 7–10 days for HAP. Central or peripheral intravenous catheter was early removed or changed in all cases of bloodstream infections to reduce the duration of antibiotic therapy at 5–7 days.

Effective antibiotics against CRE and CPE remain very limited in Italy. Actual therapeutic strategies are to increase the doses of carbapenem, colistin, and tigecycline or to combine these drugs. Double-carbapenem therapy was adopted as a salvage therapy for critically ill patients with CRE or CPE infections [19,20], but the evidence about this therapeutic option is low [21,22]. Ceftazidime/avibactam and ceftolozane/tazobactam could be useful in the management of carbapenem-sparing programs in settings with a high prevalence of ESBL-producing *Enterobacterales* and DTR Gram-negative infections.

Our study has some limitations: it was a retrospective observational non-comparative single-center study with a small sample size. The observational period was short. The cohort we present is quite heterogeneous. The low number of carbapenem-resistant infections did not allow us to evaluate the microbiological response of these cases to treatment with ceftazidime/avibactam.

However, we observed that treatment with ceftazidime/avibactam was effective in all 27 infections by microorganisms that showed resistance to meropenem, with the negativity of the microbiological analyses at follow-up. Another limitation of this study design includes the lack of a control group and the potential effects of unmeasured data. Due to its retrospective nature, detailed data on patient characteristics were not systematically collected.

It will be important to extend this study to involve more centers to enroll more patients and to be able to analyze microbiological and clinical healing. It will also be important to relate these outcomes to clinical and laboratory parameters to define their effectiveness according to the different clinical conditions and thus improve the appropriateness of the use of ceftolozane/tazobactam and ceftazidime/avibactam within the antibiotic stewardship.

## 4. Materials and Methods

We carried out a retrospective single-center observational study of microbiological isolates in adult patients hospitalized in an Italian hospital who received ceftolozane/tazobactam between May 2017 and December 2019, and ceftazidime/avibactam between May 2018—the date on which the clinical use of the drug in Italy was approved—and December 2019. We considered all the microbiological isolates from blood, urine, and peritoneal fluid cultures in patients with bloodstream infections, cUTIs, and cIAIs. Each isolate was identified by Matrix-Assisted Laser Desorption Ionization Time-of-Flight (MALDI-TOF) by VITEK^®^ MS (bioMerieux) [23,24,25,26]. Antibiotic susceptibility testing was performed by Card AST-N376 and N397 by the VITEK^®^ 2 instrument (bioMerieux). Carbapenemase-producer *Enterobacterales* (CPE) strains were confirmed by microdilution (Sensititre™, Thermo Fisher Scientific) EURGNCOL and DKMGN plates. Phenotypical CPE resistance was confirmed by synergic test diffusion Diatabs™ (Rosco Diagnostica) on Muller-Hinton agar (Vacutest-Kima). The genotyping test for CPE resistance was performed by RT-PCR (GeneXpert^®^) with Xpert^®^ Carba-R test (Cepheid Inc.). The MIC values were interpreted according to the current European Committee on Antimicrobial Susceptibility Testing (EUCAST) clinical breakpoints.

Patients who received antibiotic treatment with ceftolozane/tazobactam and ceftazidime/avibactam for ≥72 h of therapy were included in the study. All the patients were followed up for at least 30 days after the ceftolozane/tazobactam and ceftazidime/avibactam therapies were discontinued. Cultures were made on blood, urine, and peritoneal fluid samples at 48 h and 72 h and 7 days after the end of antibiotic therapies. Microbiological cure was defined as culture analyses that resulted in being negative at 72 h and 7 days after the end of the ceftolozane/tazobactam or ceftazidime/avibactam therapies.

Infections were defined as the isolation of GNB classified as MDR, XDR, PDR, or ESBL-positive *Enterobacterales* infection. We also considered sepsis as a severe infection, defined as life-threatening organ dysfunction caused by a dysregulated host response to infection [27] and blood culture positivity, without a definite organ or system involvement. Ceftolozane/tazobactam was administered at the standard dosage of 1 gm/0.5 gm IV q8 h and ceftazidime/avibactam was administered at the standard dosage of 2 gm/0.5 gm IV q8 h. Dosage adjustments were made according to creatinine clearance.

The drugs administered in this observational retrospective study were used according to the technical data sheet and indications of the Italian drug agency (Agenzia Italiana del Farmaco AIFA).

In cases of off-label use, the informed consent of the patient was recorded in hospital medical records in writing. All the national laws in force at the time of data collection were respected. The appropriateness of ceftolozane/tazobactam and ceftazidime/avibactam was defined as when these antibiotics were used according to specific indications approved by our local regulatory organizations. The usefulness was defined by the results of microbiological cure according to the indications and dosage. First-line antibiotics were defined as antimicrobial agents other than ceftolozane/tazobactam or ceftazidime/avibactam used in the first step of therapy. Therapeutic failure was defined as the persistence of positive culture analyses after 72 h of antimicrobial treatment.

### Statistical Analysis

Descriptive statistics were applied to the collected data and the related results were reported as numbers and percentages for categorical data and medians and interquartile ranges (IQRs) for continuous data.

Categorical variables were compared using the χ^2^ or Fisher exact test when appropriate.

## 5. Conclusions

In conclusion, this observational study showed the high microbiological cure rates of ceftazidime/avibactam and ceftolozane/tazobactam for treating severe infections caused by MDR-GNB other than CRE and CPE, in accordance with the literature data [28,29,30,31].

Further studies are needed to provide high-quality evidence and guide the selection of effective strategies to treat MDR-*Enterobacterales*, also considering the novel β-lactamase inhibitors vaborbactam and relebactam and the next-generation antibiotics such as cefiderocol [32,33,34,35], aminoglycoside plazomicin [36,37], and tetracycline antibiotic eravacycline [38,39,40,41,42], which could be alternative therapeutic options for CRE and CPE infections [43].

Microbiological stewardship and source control represent other pillars for an appropriate use of antibiotics and to reduce the spread of resistance to antimicrobials.

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
