# Peer review of "Ceftolozane/Tazobactam and Ceftazidime/Avibactam for Multidrug-Resistant Gram-Negative Infections in Immunocompetent Patients: A Single-Center Retrospective Study"

_antibiotics, 2020, doi:10.3390/antibiotics9100640_

Round 1
Reviewer 1 Report
Major Comments
- The primary objective listed as microbiologic cure (lines 73-74), this definition is not described in the methods. The secondary objective of “appropriateness of ceftolozane/tazobactam and ceftazidime/avibactam antibiotics in GNB infections” (lines 74-75) also does not have a definition detailed in the methods. These definitions need to be included. The only definition listed in the methods is for clinical outcomes, which is not a listed objective in the introduction. If this definition of “infection cleared with no positive cultures at the end” of antibiotics is describing microbiologic cure, this needs to be clearly stated. If using this definition, did all patients have repeat cultures to test for culture clearance? This should be clearly stated in the methods, results, and limitations. A percentage of clinical cure (line 99, 125) is listed for both the ceftolozane/tazobactam and ceftazidime/avibactam groups, though this is not listed as an objective and no definition is provided.
- Line 100-103: “Microbiological analyses were shown to be negative in 219/241 samples.” These results are unclear. Does this imply that the majority of patients did not have bacteria isolated? However, this does not align with the organisms listed as being isolated in lines 88-90. Are these repeat isolates? This N could possibly be better represented in terms of number of patients that cleared their culture (initially positive, with negative repeats), though this definition is not stated anywhere. The ceftazidime/avibactam microbiological cure is reported with patient number as N, would be consistent and this denominator makes more sense.
- Line 124: The authors state “there was no association between clinical failures and type of primary infection.” This is a broad conclusion to make with no data presented to support this statement.
- Line 163-175: This portion of the discussion section includes information that is better suited for the results as it contains new information. These two paragraphs state that statistical significance was reported for various organisms and antibiotics, however it is unclear what was evaluated and what is significant. It is unclear what is being compared. The methods and statistical analysis descriptions need significant clarification and the results of this should be moved to the results section, rather than discussion.
Minor Comments
- There are some statements that warrant citations, with examples including:
- Line 44-45: authors remark that MDR GNB infections are responsible for prolonged hospitalization – need to reword that this is a potential or cite
- There are some sentence that require clarification or rewording for better understanding. Examples include:
- Line 42-43: Should be reworded for improved understanding; can be more simply stated that MDR GNB are difficult to treat
- Line 45-48: The authors state there are increasing infections caused by various organisms, though the verbiage utilized doesn’t get across what I believe the intent to convey is – more resistant infections with less treatment options leads to increased morbidity and mortality; depending on how the sentence is altered, may require citation
- Line 49-52: The intent of this sentence is unclear; as written now, I interpret to mean that gram negative infections that are NOT MDR/XDR/PDR’s require second-line agents
- Line 60: Ceftazidime/avibactam is also FDA/EMA approved for the treatment of HABP/VABP, would list for completeness
- Line 66-67: Ceftolozane/tazobactam is also approved for HABP/VABP
- Line 82-83: If inclusion required at least 72 hours of ceftolozane/tazobactam, this should be listed in the methods; additionally, this sentence states 241 ESBL-Enterobacteriales isolates were included, however the methods allow for additional organisms to be included (i.e. Pseudomonas, which was isolated per line 88); Line 89-90 states that ESBL-producing Enterobacteriales were 136/241 (56.4%), which is confusing given the first sentence.
- Line 82-90: The total isolates is 241 for 122 total patients. This suggests that not all patients had a repeat culture obtained, can microbiologic cure be determined? Are negative cultures not included in this N? Rather than describing the source in terms of patient condition, may be clearer to describe in terms of culture source (urine, blood, peritoneal fluid, etc.). Did patients listed as ‘sepsis’ have positive blood cultures?
- Clinical appropriateness, listed as an objective, is not described in the ceftolozane/tazobactam results.
- Would be relevant to include duration of antibiotic therapy with ceftolozane/tazobactam and if/when the cultures were collected in relation.
- Line 107-108: If inclusion required at least 72 hours of ceftazidime/avibactam, this should be listed in the methods; additionally, this sentence states “for GNB infections other than CRE” which is misleading given the statement in the methods that MDR, XDR, PDR, and ESBL positive infections were included. Based on this sentence, it is understood that CRE is excluded.
- Line 122-124, 127-128: Given the frequency of combination therapy, would include the n for each combination agent, rather than just list them. Microbiological cure for combination therapy is listed as 69.9%. Were there any relevant combinations that should be listed as significantly more or less successful?
- Line 130-131: Would state no adverse events related to ceftolozane/tazobactam in the ceftolozane/tazobactam section, separate from the ceftazidime/avibactam statement given the rest of the data has been grouped and described separately.
- 30-day mortality is evaluated. Please state is this is ‘all-cause mortality’ or describe how otherwise determined.
- Line 148-149: It is stated that appropriateness of use of these drugs are assessed, however, this is not defined in the methods, nor reported in the results.
- Polymicrobial infections need to be discussed. Based on the number of each type of organism identified, it is suggested that there are patients with more than one organism identified, however, this is never stated. These cases need to be identified and addressed.
- Authors need to address the varying clinical syndromes. Sepsis is not defined. The clinic syndromes included each require varying approaches regarding source control, treatment duration, and criteria for determining clinical improvement or cure. Given the large amount of patients with intraabdominal infections, it would be prudent to explicitly address outcomes in this patient population.
- Clinical utility would be significantly improved if the authors specified the patients with carbapenem resistant organisms.
- Table A3/A4: The “total of antibiotics” section is unclear. Does this include all organisms? Why are the N’s varied? The numbers within the chart are not correctly adding up. Abbreviation for column 3 (N. TOT) not defined. ESBL should not be listed as an antibiotic.
- Many references missing journals and/or dates. Examples include reference 4 missing journal title, reference 11 missing journal and date of publication.
Reviewer 2 Report
Manuscript - Ceftolozane/tazobactam and ceftazidime/avibactam for multidrug-resistant Gram-negative infections in immunocompetent patients: a single-center retrospective study has been evaluated for publication in Antibiotics and publication of the paper is proposed after a major revision.
In the present study, ceftolozane/tazobactam and ceftazidime/avibactam were used in a case study for the treatment of severe Gram-negative infections in Ferrara, Italy, for the University-Hospital period 2017 to 2019.
Comments: The authors are suggested to improve:
Lines 27 - 35, (abstract): Abstract is written too general and should be improved. The novelty of the work is missing.
Lines 78 - 187 (Results and Discussion): All available data are presented in 4 tables, therefore it is suggested that the authors replace some tables and add some figures or present the data in a different form.
Lines 183 – 187 (Discussion): The authors claim: »Our study has some limitations: it was a retrospective observational non-comparative single center study with a small sample size. The observational period was short. Cohort we present is quite heterogeneous. Another limitation of this study design includes the lack of a control group and the potential effects of unmeasured data. Due to its retrospective nature, detailed data on patient characteristics were not systematically collected.« I agree with this statement, but the authors should explain their future plans and improvements and discuss this with literature data on similar studies outside Italy. It is proposed to add a discussion similar to the one in the first paragraph in the conclusions.
Line 210, (Materials and Methods), Data for MALDI-TOF analysis missing - insert data or add references.
Lines 260 - 280 (Appendix), in Tables 1 and 2 a uniform number of decimal places is not used, some data are presented without, some with one and some with two decimal places.
Round 2
Reviewer 1 Report
Attached are my comments. I believe that the authors still need to clarify parts of their research. My responses to the authors are in RED.
Reviewer 2 Report
The authors have improved the work considerably and accepted all comments, so I propose to publish the work in the present form.Author Response
September 15, 2020
Sir,
I would like to thank you very much for your comments that allows us to improve our manuscript.
Best regards.
Giovanni Gabutti.
Round 3
Reviewer 1 Report
No Comments